# Interpretable Passive Multi-Modal Sensor Fusion for Human Identification and Activity Recognition

**DOI:** 10.3390/s22155787

**Published:** 2022-08-03

**Authors:** Liangqi Yuan, Jack Andrews, Huaizheng Mu, Asad Vakil, Robert Ewing, Erik Blasch, Jia Li

**Affiliations:** 1Department of Electrical and Computer Engineering, Oakland University, Rochester, MI 48309, USA; liangqiyuan@oakland.edu (L.Y.); jackandrews@oakland.edu (J.A.); huaizhengmu@oakland.edu (H.M.); avakil@oakland.edu (A.V.); 2Sensors Directorate, Air Force Research Laboratory, WPAFB, Dayton, OH 45433, USA; robert.ewing.2@us.af.mil; 3Information Directorate, Air Force Research Laboratory, Rome, NY 13441, USA; erik.blasch.1@us.af.mil

**Keywords:** human identification, activity recognition, sensor fusion, passive radio frequency, passive infrared

## Abstract

Human monitoring applications in indoor environments depend on accurate human identification and activity recognition (HIAR). Single modality sensor systems have shown to be accurate for HIAR, but there are some shortcomings to these systems, such as privacy, intrusion, and costs. To combat these shortcomings for a long-term monitoring solution, an interpretable, passive, multi-modal, sensor fusion system *PRF-PIR* is proposed in this work. *PRF-PIR* is composed of one software-defined radio (SDR) device and one novel passive infrared (PIR) sensor system. A recurrent neural network (RNN) is built as the HIAR model for this proposed solution to handle the temporal dependence of passive information captured by both modalities. We validate our proposed *PRF-PIR* system for a potential human monitoring system through the data collection of eleven activities from twelve human subjects in an academic office environment. From our data collection, the efficacy of the sensor fusion system is proven via an accuracy of 0.9866 for human identification and an accuracy of 0.9623 for activity recognition. The results of the system are supported with explainable artificial intelligence (XAI) methodologies to serve as a validation for sensor fusion over the deployment of single sensor solutions. *PRF-PIR* provides a passive, non-intrusive, and highly accurate system that allows for robustness in uncertain, highly similar, and complex at-home activities performed by a variety of human subjects.

## 1. Introduction

With the urgent need for smart living and hazard prevention, there currently exists a need for an accurate, inexpensive, and non-invasive at-home monitoring solution. Towards this end, the field of human identification and activity recognition (HIAR) can be presented. HIAR has applications in a variety of medical domains, including elderly monitoring [1,2,3], smart living [4], and medical care [5]. Currently, HIAR technologies are primarily deployed via computer vision [6], wearable sensors [7], and ambient sensing [8] methods. Computer vision systems for HIAR provide shortcomings that deter application for a long-term deployment solution by their end-users. One of these shortcomings is the intrusive nature of its design, causing privacy concerns to the monitored subjects. Wearable sensor systems are generally composed of one single inertial measurement unit (IMU) or combined with many diversified sensors to perform HIAR [9]. Wearable sensors are also unsuitable for long-term monitoring due to their uncomfortable characteristics and the requirement to remember to wear terminal equipment, which is often an issue in neurodegenerative patients and elderly populations. Fortunately, because the data collected by ambient sensors only represents environmental changes, these sensor systems are advantageous for monitoring solutions as they are non-intrusive, accurate solutions for HIAR [10,11,12].

HIAR classifications via ambient sensing have commonly been performed by passive infrared (PIR) sensors that are deployed in the occupied indoor space [13,14,15,16]. PIR sensors are inexpensive, commercial-off-the-shelf (COTS) components that detect infrared radiation in their field of view (FoV). PIR sensors are often used as motion detectors, as their internal pyroelectric elements will detect the voltage differences that result from the movement of a human subject across its FoV, triggering a positive result. Due to this reliance on infrared motion, PIR sensors cannot accurately and reliably detect stationary human occupants. To combat the stationary detection problem that adversely affects PIR sensors, a motion-induced PIR sensor (MI-PIR) was proposed previously and has been shown to be accurate for stationary human detection, HIAR, and other regressions and classifications in multiple ambient environments using only one PIR sensor modality [17,18]. As MI-PIR had proven to be successful, expanding the system in this work to detect both stationary and non-stationary targets in a proposed passive multi-modal sensor fusion system seemed to be the next logical step towards increased robustness.

Leveraging ambient signals for HIAR classification extends beyond the PIR sensing technology. Similar to PIR, passive radio frequency (PRF) is also an ambient sensing technology that passively acquires environmental signals. The passive signal acquisition method of PRF has attracted our attention because of its non-intrusive, pollution-free, and relatively inexpensive advantages. Software-defined radio (SDR), a wireless communication system that combines software and hardware, is widely used because of its convenience, flexibility, and anti-interference characteristics [19,20,21]. In our research [22,23,24], the SDR control module scans the human-sensitive frequency band to collect PRF data for HIAR classification. The realization of our PRF technology does not require any transmitter or wireless signal transmission frameworks such as Wi-Fi or a cell network. Previous work demonstrated that using the SDR device to collect PRF information is an accurate method for human detection, showing greater than 0.90 accuracy for ten out of the overall twelve frequency bands tested for in [22]. The most significant limitation of this research is that the SDR device and PRF information is susceptible to external interference, such as metal cabinets in an indoor environment. For this reason, we propose a sensor fusion approach using PIR and PRF data.

Due to the complex nature of both raw MI-PIR voltage data and PRF information, deep learning was utilized to learn from the slight variations that exist between the various scenarios found within the dataset [17,18,22]. Deep learning, in comparison to traditional machine learning, can perform automatic feature extraction and thus has advantages in a HIAR context [25]. Since HIAR tasks require powerful time-series processing capabilities, recurrent neural network (RNN) models are used for the realization of HIAR classification tasks. RNNs have shown success in a variety of additional human recognition applications such as gesture recognition [26], gait analysis [27], and translation [28]. The addition of long-short term memory (LSTM) units in the RNN model serves to remove the vanishing gradient issue that often plagues RNN frameworks with increasing epochs [29]. Due to the success of this deep learning model, as well as due to the fact that MI-PIR data and PRF information are both temporal in nature, the RNN model framework with LSTM units is proposed in the *PRF-PIR* system for accurate HIAR classification.

There exist numerous types of single modality and single-type ambient sensing solutions reliant on deep learning algorithms in literature for accurate HIAR. In these instances of single modality and single-type sensor solutions, there exist shortcomings in the case of only deploying one sensor, as well as the shortcomings based on the chosen sensor itself [30]. For example, low recognition accuracy can persist in the case of similar activities, and specific sensor solutions have shown to be susceptible to external interference limitations such as different environmental temperatures and electronic interference. There also exist demerits based on the modality chosen, such as in the instance of PIR sensors being unable to detect stationary human subjects reliably and accurately in the traditional sense. In the specific case of MI-PIR, the system is adversely affected by the dependence on motion, e.g., classifications have a much larger dependence on the ambient environment due to the induced motion. In the case of SDRs, classification accuracy is adversely affected by electronic interference, where HIAR accuracy is dependent on the location of the SDR device antenna deployed. To combat these mentioned issues for each sensor system and to provide a highly accurate HIAR system that is suitable for a long-term human monitoring solution, we propose *PRF-PIR*, a sensor fusion system that learns from the PRF information collected by the deployed SDR device and the infrared radiation collected by the novel MI-PIR system. *PRF-PIR* utilizes an RNN deep learning model with LSTM units to learn the complex scenarios that exist in the multiple data streams. The proposed sensor fusion approach to HIAR was validated via the data collection of twelve subjects completing eleven different activities. One of the eleven activities was a simulated fall event, which is explicitly included to highlight the efficacy of the sensor fusion framework for an at-home, independent, long-term human monitoring solution.

Finally, as the initial predictions are made using a black-box system, we provide transparency and interpretability using decision-level fusion and SHAP (SHapely Additive exPlanations). For the purposes of this paper, interpretability refers to the connection between the data input to AI and its predictions, while transparency refers to the ability of human users to understand the decision-making process of the algorithm used. For the decision-level fusion model itself, the use of SVM (Support Vector Machines), a linear supervised learning method, adds further transparency. The contributions of this work are as follows:An interpretable, passive, multi-modal, sensor fusion system *PRF*-*PIR* for human monitoring is accurate at HIAR and is non-intrusive, transparent, passive, and inexpensive in design. To our best knowledge, this is the first passive sensor fusion system for human monitoring applications.The proposed system mitigates the limitations of single modality solutions, such as the vertical FoV and ambient dependence on MI-PIR and the impact of electronic interference on PRF. *PRF-PIR* provides a robust, high-accuracy, and reliable classification system for the HIAR task.In addition to the transparent nature of the decision-level fusion, SHAP is used to interpret how the system reduces the influence of vertical FoV, ambient dependence, and electronic interference and provides a visual application prospect.

This paper is structured as follows. In Section 2, a background is provided on the modalities and related topics referenced in this work. Section 3 explores the methodology of the proposed system, including for the two single-type of sensors, the fusion method, and for the RNN architecture. Section 4 shows the completed experiments and the results obtained. Before summarizing this research in Section 5, limitations and future work are explored in Section 6.

## 2. Related Work

This section first highlights the related work using SDR to collect PRF information and the PIR sensing solutions for human monitoring applications. The technical backgrounds of each modality and other related topics are briefly introduced in this section. Previous research utilizing the SDR devices to collect PRF information and developing PIR systems to detect stationary human subjects will be highlighted to aid in the technical backgrounds and related work of these modalities. Solutions using sensor fusion for human monitoring applications and those solutions providing XAI rationale will be introduced last.

### 2.1. SDR/PRF

Device-free HIAR based on RF has been an emerging field in recent years. HIAR based on RF signals, commonly focuses on RF identification (RFID) [31]. RFID has many advantages for HIAR, such as its passive nature, minuscule size, and lack of battery requirement. Currently, the common method for device-free HIAR is to use external RF sources (fixed RFID tags can be seen as transmitters) to transmit the RF signal, and the reader receives the RF signal reflected by the human body [32,33,34]. Although these methods avoid the trouble of wearing RFID tags, they require humans to perform activities in areas covered by RF signals. In this research, a pure device-free PRF sensor technology based on the SDR does not require RFID tags or other transmitters. As a result of no tags or transmitters required, the SDR technology is seen as an advantageous solution for HIAR applications, and specifically, a human monitoring applications.

Previous work has leveraged the SDR device to collect PRF information for human occupancy detection in different scenarios such as study rooms, bedrooms, and vehicles [35]. Likewise, six SDR devices were deployed in different locations to solve the task of human indoor positioning [23]. Four machine learning models—support vector machine (SVM), k-nearest neighbors (k-NN), decision tree (DT), and random forest—proved their potential in processing PRF data. An RNN model will be presented in this paper as a technology for processing PRF data because of its excellent ability to predict time-sequential tasks, such as for HIAR.

### 2.2. PIR Sensor

PIR sensors accurately detect human subjects when a change in infrared radiation exists. Since PIR sensors are dependent on this infrared radiation change, there have been numerous solutions proposed in the literature to solve this issue for accurate stationary human detection with a FoV reliant on the Fresnel lens specifications. For reference, Fresnel lenses are modules that sit at the edge of the PIR sensor and serve to expand the FoV of the sensor into many alternating positive and negative zones [36]. Libo Wu, Ya Wang, et al. proposed numerous solutions for solving the stationary human detection problem with a single PIR sensor modality [37,38,39]. These proposed solutions utilize a liquid crystal shutter that periodically chops the long-wave infrared radiation to create the necessary infrared radiation change artificially. A sensing module dependent on an optical shutter was also designed and proposed to solve the stationary human detection problem in [40]. This work shows an accurate detection of standing and moving human subjects at a maximum distance of about seven meters. At a maximum distance of one meter, a traditional analog PIR sensor, coined CM-PIR, was introduced in our previous work to solve the stationary human detection problem. CM-PIR proved 0.94 accurate at differentiating perfectly human subjects from unoccupied scenarios by detecting the movement of the chest of the monitored subject [41]. As mentioned in [17,18], the motion nature of MI-PIR was used to expand the normal FoV of the Fresnel lens from a manufacturer reported 93 degrees to a FoV of 223 degrees via an induced 130-degree rotation. From this research, it was experimentally determined that the maximum sensing distance of the MI-PIR system is greater than 21 m.

PIR sensor solutions for HIAR have also been explored in the literature. For example, the work presented [40] also extends their stationary human detection solution to activity recognition, proving 0.93 accurate at differentiating unoccupied, sitting, and walking. The work presented [42] highlights an activity recognition F-measure of 0.57 while using two traditional PIR sensors to recognize four complex activities completed at the same location. By expanding the FoV and maximum sensing distance and providing accurate stationary human detection and HIAR classifications, the MI-PIR system is seen as a superior solution. In this work, we expand the capabilities of the solution by further increasing data collection complexity and reducing the limitations of the MI-PIR system via the inclusion of the SDR device.

### 2.3. Sensor Fusion

Sensor fusion is a common approach to complementing sensor modalities for HIAR [1]. Bazo et al. proposed a sensor fusion model, Baptizo, in [43] that it leverages active RF positioning data captured with ultra-wideband (UWB) devices and RGB-Depth (RGBD) human pose estimation for the reduction of human positioning error to assist with the eventual activity recognition classification. This work can be applied to clinical environments, where a human subject may be behind an occlusion in a tight environment and unseen by the RGBD, allowing for sensor fusion to enhance the robustness and classification accuracy. In addition to this work, there exist various works that propose sensor fusion to accurately classify fall events, which is applicable for an at-home monitoring solution. In [44], a smart fall detection framework is proposed that complements video and ultrasonic sensors for accurate and faster fall detection. The human motion features from the video and the longitudinal motion signals from the ultrasonic sensors are combined to develop a three-dimensional (3D) movement trajectory map that achieves 0.90 accuracy in the detection of normal, transition, and fall motions. Sensor fusion and transfer learning was proposed in [45] to combine multiple radar sensors for accurate activity recognition and fall detection. One frequency modulated continuous wave (FMCW) radar and three UWB radars were combined to achieve 0.84 accuracy of twelve different activities using the VGG-16 convolutional neural network (CNN) model and hard fusion with the Naïve Bayes combiner (NBC). The works presented in the [43,44] present video modalities are intrusive to the monitored subjects, and the work presented in [45] presents a solution that utilizes active RF sensors that come with negative energy and health concerns. Therefore, a non-intrusive, inexpensive, pollution-free, and accurate monitoring solution can fill the gaps in the systems introduced in the literature.

Many sensor fusion solutions have flourished in recent years due to their accuracy, reliability, and robustness [46,47,48,49]. Sensor fusion is traditionally described and divided into the data-level, feature-level, and decision levels [49] based on when fusion is implemented. Traditional fusion methods and feature extraction usually are statistical information such as average value, standard deviation, power, energy, and entropy in the time domain and frequency domain. Although these kinds of handcrafted feature information extraction methods are feasible in simple activities, for complex applications with a large number of potential variables, such as HIAR, automatic feature extraction such as deep learning has received the attention of researchers. Therefore, direct data-level fusion is used as in our sensor fusion method, and the work of feature extraction is delegated to RNN for processing. Decision-level fusion belongs to the high-level data level, having processed and predicted more clearly at a higher level of data processing. Related research [50] had previously implemented decision-level fusion by considering the predictions via the confusion matrix for each input. This approach is useful when the model is unable to take into account the temporal aspect of the data being used, but for the purposes of this paper, the RNN architecture mitigates that benefit and allows us to reduce the complexity of our decision-level fusion model.

Although both the SDR device and PIR sensor have their shortcomings, complementary sensor fusion combined with the deep learning RNN model to achieve complex HIAR is proposed and implemented. To the best of our knowledge, this is the first paper that leverages PRF information and PIR sensor data through sensor fusion for proposed human monitoring.

### 2.4. Explainable AI (XAI)

XAI is a relatively new and emerging field in machine learning. Fundamentally, XAI serves to provide explanations to models previously seen as black-box and indicate how the data affects the model output. Despite XAI’s emergence and various applications in machine learning, there has yet to be a widely adopted standard, let alone a widely adopted method of quantifying explainability for explaining models. Even simply discussing methods of quantifying such approaches is quite a task, as there are several different classifications when it comes to types of explainability. Before we can even begin discussing the overall topic of XAI, the most important thing to do first is to define explainability (or interpretability) itself.

For different classifications, the definition of explainability is necessary. The operation structure of the black-box model and which data features affect the model output is the focus of related work. Explainability might be defined as mapping the predicted class into a domain that the human user might be able to make sense of, such as the feature difference between image pixels of cats and dogs, i.e., the process of machine translation tasks. In an ideal system, one might even define explainability as a reasonable explanation for why a collection of features contributed to the decision-making process or at least determining how much weight the decision-making process gave to said features [51]. From saliency maps to activation maximization, there are a few methods by which explainability can be achieved.

On the other hand, interpretability provides an explanation of the link between model input and output, with similar predictability. The distinctions between these types of methods are typically twofold, described as either Ante-hoc or Post-hoc, local or global, or model-specific or model agnostic [52]. Ante-hoc systems provide explanations from the beginning of the model. These systems enable one to gauge how certain a neural network is about its predictions [53]. Post-hoc techniques entail creating interpretability into a model based on its outcome, marking the part of the input data responsible for the final decision. Like Ante-hoc techniques, this also can include visualization and saliency mapping but also uses methods such as gradients or feature importance. The definition of explainability (or interpretability) is not absolute but depends on the needs of users. For our sensor fusion system, the advantages of sensor fusion over single-type sensors need to be interpreted urgently.

For the purposes of this paper, we use SHAP [54,55], an extremely versatile game-theoretic algorithm that works with most black-box machine learning algorithms to understand a data set’s impact on a model. For the proposed sensor fusion system used to implement HIAR applications, we are more concerned about the relationship between the input data and the predicted results rather than a black-box model process. Therefore, the interpretability of the system, rather than explainability, has gained our attention. We take a more post-hoc approach towards providing interpretability, using decision-level learning to replicate the process. According to our needs, the information provided by decision-level learning is more related to activities than data or features, so the insights provided allow us to infer the behavior of the fusion model. In future work, the use of other methods, such as counterfactual methods or rationalization, may provide better insight into the model’s behavior.

## 3. Methodologies

The proposed *PRF-PIR* system consists of one SDR device and one novel MI-PIR sensor system for the data collection of various activities completed by many unique individuals in an academic office space. The methodologies of each modality and the sensor fusion process will be explained in this section.

### 3.1. Data Acquisition

This subsection will provide detailed methods pertaining to the two ambient sensors utilized in this work: the SDR device and the MI-PIR sensor system.

#### 3.1.1. SDR Device

Following the determination of the human body sensitive frequency to be 330 MHz in [35], the scanning frequency band B of the SDR device is set to 300 MHz to 420 MHz. The SDR sampling rate is set to 2.4 MHz to obtain more detailed data. Although the sampling time of SDR is about eight seconds per sample, three samples are combined into one to ensure the commonality of sample length in respect to the PIR sensor data. The average power subtracted from the mean and amplified 10 times is used as the PRF data:(1)$$P\left(f\right)=\left[10\log_{10}\frac{\sum_{i=1}^{N}{\left(\frac{s_{f}\left(i\right)}{127.5}-1\right)}^{2}}{N/2}-M\right]\times 10,\;\mathrm{for}f\in B,$$
where average signal power *p* is a function of the frequency band center *f*, *N* is the number of samples in each frequency band, which is 4800 in our experiment, $s_{f}\left(i\right)$ is the value of the *i*-th raw data received by the SDR device when the frequency band center is *f*, and *M* is the average value of the PRF data set. The average signal power *p* is subtracted from the *M* to eliminate the bias in the PRF data set and amplify it by ten times to enlarge the difference between human-sensitive data and noise. Figure 1 shows the SDR device antenna used to collect PRF data.

#### 3.1.2. PIR Sensor

The MI-PIR system utilized in this work is composed of an Elegoo Uno R3 microcontroller, a Dynamixel MX-28 robotic actuator, a flat platform, a thermal insulator, and an analog capable Panasonic AMN 24112 PIR sensor. The robotic actuator rotates the analog PIR sensor in a 130-degree motion every 36 s. The first 26 s rotate the PIR sensor forward, and the next ten seconds rotate the PIR sensor back to its original position. Since a rotating PIR sensor will also detect most stationary objects, the analog information from the PIR sensor is extracted to learn from the slight voltage variations collected from the indoor environment. The microcontroller utilized in this system serves to transfer the analog information from the sensor to the Arduino integrated development environment (IDE) for extraction and processing. The thermal insulator, on the other hand, serves to eliminate some of the nearby infrared that radiates from PCs and monitors that are in close proximity. A more long-term material for thermal insulation is left for future work. Moreover, the flat platform of the MI-PIR system serves no other purpose than to provide a stable surface for the thermal insulator to sit. The PIR sensor in (a) and the complete MI-PIR system in (b) are presented in Figure 2.

The PIR sensor utilized in the *PRF-PIR* solution has a manufacturer-defined horizontal FOV of 93 degrees and vertical FoV of 110 degrees, with a selected sampling rate of 10 Hz. With each complete scan being 36 s at a 10 Hz sampling rate, one large file is batched into 360 data features based on each specific identification or activity label. From [17], it was determined that the signal power was a strong indicator of human presence in the MI-PIR system. Signal Power is calculated as the absolute value of the fast Fourier transform (FFT) coefficients and is represented below in (Equation 2).
(2)$$\mathrm{Signal}\;\mathrm{Power}=\left|\int f\left(t\right)e^{-j\omega t}dt\right|.$$

### 3.2. Sensor Fusion

After sensor fusion is performed at the data level for PRF and PIR, the predictions go to the decision fusion. The *PRF-PIR* model presents the predictions to realize the interpretability, which is the structure of our system shown in Figure 3. Owing to limitations with the data structure of the PRF and PIR data, decision-level fusion became necessary to provide transparency.

Sensor fusion at the data level is one of the primary methodologies of our *PRF-PIR* system. PRF and PIR have different feature vectors in terms of physical meaning, unit, sampling period, the number of features, etc. According to the time required of the SDR device to scan the human body’s sensitive frequency band and the time required to complete one full MI-PIR scan, multiple pieces of PRF information are combined to adapt to the time period of the PIR sensor. Following the SDR and PIR modalities data collection method in Section 3.1, the *PRF-PIR* data set is generated as a fusion data set of data-level. The feature vector of PRF-PIR sensor fusion has 663 data features obtained by concatenating 303 data features of PRF and 360 data features of PIR. The extraction of the data features benefits from the automatic calculation of the weight and bias of the LSTM unit in the RNN. This work focuses on multi-modal sensor fusion, a straightforward RNN model architecture as a classification tool for preliminary validation in the experiments. Three RNN models with the same architecture are developed on the PRF, PIR, and *PRF-PIR* data sets, respectively. The same RNN model architecture can intuitively reflect the feature containment capability of different sensors in implementing HIAR tasks. Therefore, the PRF, PIR, and *PRF-PIR* data sets are trained by the same RNN model architecture to obtain the predicted output of PRF, PIR, and *PRF-PIR* sensor fusion. After RNN training, these three predictions are created into a decision-level fusion data set, which is trained by the decision-level fusion model. SHAP XAI is used to realize the interpretability of the decision-level data set, which is the impact of the three predictions on PRF, PIR, and *PRF-PIR* sensor fusion on HIAR categories.

The developed RNN model architecture has five layers, as identified in Figure 3. Before inputting the time serial data into the LSTM, a dense layer is used to reduce the number of features. After LSTM processes the time serial data, the dimension is reduced to the number of classification categories through two dense layers. According to the different input data sets and classification tasks, the input and output Shapley values of the RNN model will change accordingly. The model is trained using the Adam optimizer and an initial training speed of $10^{-3}$ with a decay factor of 0.1 times every one hundred epochs. The results of training RNN on PRF, PIR, and *PRF-PIR* sensor fusion datasets are presented in Section 4.2, Section 4.3 and Section 4.4.

## 4. Experiment and Results

The proposed *PRF-PIR* sensor fusion framework is verified via the data collection of eleven activities completed by twelve different human subjects. The data collection process was completed in an indoor office space on the campus of Oakland University. This section will highlight the data collection and the corresponding results of the *PRF-PIR* system.

### 4.1. Experimental Set-Up

Data collection was completed in the academic office environment, which is illustrated below in Figure 4. The experimental data collection process followed the principle of diversification and is close to the daily activities of student researchers. The office environment highlighted in Figure 4 is 5.18 m in length and 3.96 m in width, which is comparable to a residential environment and an adequate size to test the capability of the *PRF-PIR* system as an at-home human monitoring application. Figure 4 illustrates the six activity locations for which the eleven activities were completed by the twelve human subjects. The six activity locations correspond to areas that are near desk locations, as well as one location (Location 6) that corresponds to the empty space at the center of the office space. The locations of Subjects 1 through 6 in Figure 4 correspond to the locations in the office space where activities were conducted. During the data collection, a subject simulates the activity in a continuous fashion at one of the six locations based on the specific activity. Five tentative SDR antenna placement locations and the MI-PIR sensor system are also included in Figure 4. Based on past experience in [23,35], electronic interference objects such as metal cabinet, refrigerator, and printer can affect the performance of SDR antennas, and due to this, these objects are also marked in Figure 4. The inclusion and placement of the five SDR devices in the office space were towards identifying the optimal placement of the SDR antenna for the *PRF-PIR* system. MI-PIR is placed near Location 1 as its optimal placement due to the higher vantage point conflicts with the vertical FoV limitation, ability to scan the entire space, and the low frequency of student usage at this site [17].

Table 1 highlights the overall dataset collected and indicates accurate diversification of activities. The Activity Recognition ID, Activity, Human Subject ID, Location, and the scenario description are included in Table 1. The twelve subjects that volunteered in the data collection all performed the “Smartphone” activity (Activity Recognition ID 1) and any additional activities that they had the availability to complete. This process ensured that the human identification classification could be completed for one unique activity. Furthermore, Human Subject ID 2 completed all the activities so that an activity recognition accuracy could be classified for only one human subject. Some of the activities were performed at varying locations to ensure the robustness of the model, as the model could potentially use the locations as the differentiators for learning and not the activities themselves. The “Smartphone” and “Board” activities were performed at specific locations due to the need for accurate human subject identification and the location constraint of the whiteboard, respectively. The same training strategy is applied to the three RNNs trained on the three data sets, respectively. In our experiments, 70% of the collected data was used to train the RNN model, while 30% of the collected data was used for testing. The training data set is only used to train three RNN models, and all results are based on an independent test data set.

In addition to Table 1, Table 2 is included as following, which shows the difference in age, physical body information, and body mass index (BMI) of the twelve subjects invited in this research. The twelve subjects included eight men and four women, all of whom were students at Oakland University. This physical information is critical to record as the uniqueness of the human subjects slightly alters the infrared and RF in the academic office space. The variations between the minimum and maximum of each category highlight the data collection, e.g., one can identify that human subjects recruited were primarily university students, and there exists some variation in height and weight.

### 4.2. Optimization of SDR Device Antenna Location

Although the SDR device can receive RF signals over a large environment, the locations of the SDR device antenna are essential to consider due to SDR’s susceptibility to interference. Table 3 highlights the accuracy of the five antenna locations and the improvement of sensor fusion in the HIAR classification task. From the accuracy recorded in the PRF column of Table 3, the interference received by SDR devices in varying locations is different. From the perspective of classification accuracy only, Antenna E received the most severe interference, while Antenna C received the least. From the classification results, the error in Antenna E is not limited to a single category but is widely distributed between each subject and activity. Therefore, the reason for the drop in accuracy is not data distortion caused by the relative position of the receiver to the subject. On the other hand, the SDR device receiving frequency is set to a human-sensitive frequency band, so interference from electrical equipment such as computers can be excluded. Therefore, the metal cabinet near Antenna E is leading to a decrease in accuracy due to the physical properties of metal to block the RF signal. Antenna E near metal objects will lack a portion of the spectrum that contains human signatures, and the information contained in the collected data features will be missing, resulting in a drop in accuracy. On the contrary, Antenna C has the highest accuracy, which is to be expected. As can be seen from Figure 4, Antenna C is not only in the middle of the scenario, but it is far away from possible sources of interference such as walls, metal objects, activated computers, refrigerator, etc. And in terms of task settings in Table 1, most of the activities are set in Location 6, which is the closest to Antenna C. Therefore, the location of the antenna is highly correlated with accuracy. These results can benefit future research.

Although SDR device antennas are susceptible to external interference, sensor fusion can increase the accuracy of HIAR and reduce the effects caused by such interferences. In particular, the PRF data set with the largest interference (Antenna E) is greatly improved with sensor fusion. The PRF data set with the least interference (Antenna C) shows comparable accuracy to that of the sensor fusion result, proving the anti-interference and robustness of our sensor fusion system. In the latter case, the sensor fusion result via the *PRF-PIR* system still aids in enhancing the classification accuracy via an increase in accuracy and greater classification robustness. From these results, Antenna C is selected as the most optimal placement of the SDR antenna.

### 4.3. Experimental Results

The experimental results of this study are divided into two major categories: human identification and activity recognition (a.k.a. HIAR). The results of both human identification and activity recognition will include the results of PRF, PIR, and *PRF-PIR* sensor fusion. We further illustrate the necessity of sensor fusion in HIAR classification in this subsection, as the limited vertical FoV and ambient dependence of PIR and the susceptibility to the interference of PRF are illuminated.

#### 4.3.1. Human Identification Results

Human identification is defined as recognizing unique subjects performing the same activity in related scenarios. Sitting in a chair while using a smartphone is one of the most common laboratory activities. It is used to prove the effectiveness of human identification for the *PRF-PIR* system. Using smartphones increases the RF signal interference due to the introduction of an electronic device, and static activities are generally more difficult to recognize, and due to this, smartphone activity was identified as a strong indicator of the effectiveness of the sensor fusion strategy for a human identification classification. Overall, twelve subjects completed the smartphone activity for the human identification classification in this study. The results of the smartphone activity labeled with an Activity Recognition ID of 1 in Table 1 are presented above in Table 3. The results from Table 3 indicate a human identification accuracy for PRF, PIR, and *PRF-PIR* sensor fusion as 0.8993, 0.9530, and 0.9866, respectively. Figure 5 below provides the confusion matrices for the human identification results for (a) PRF, (b) PIR, and (c) *PRF-PIR* sensor fusion. The results of the confusion matrices indicate the effectiveness of sensor fusion at removing some confusion of identification labels and increasing the accuracy and robustness of identification in the presence of interference and static activities.

#### 4.3.2. Activity Recognition Results

Compared with the human identification classification, activity recognition is a more challenging and even more practical classification, especially in terms of human monitoring. For activity recognition, eleven activities were completed by twelve subjects overall, including the smartphone activity previously utilized for human identification. The activity recognition dataset was used to train the RNN model to achieve the classification results. As with human identification, the RNN model framework was presented earlier in Section 3.2 is utilized for classification. The results of activity recognition are presented above in Table 3. The results from Table 3 indicate accuracies for activity recognition as 0.9434, 0.8066, and 0.9623 for the PRF, PIR, and *PRF-PIR* sensor fusion framework, respectively. Figure 6 presents the confusion matrices for these three reported accuracies.

For diversified human activities, the MI-PIR sensor system’s vertical FoV limitation and ambient dependence provide multiple errors and low precision. As shown in Figure 6b, the errors of the RNN model trained on the PIR data primarily occur in the “Unoccupied” and “Fall” labels. Due to the vertical FoV limitation of the MI-PIR sensor system at an elevated location, the confusion between these two labels is heightened, as the MI-PIR system cannot accurately locate a human subject lying on the ground of the academic office space. The ambient dependence of the MI-PIR system is also proven with the “Unoccupied” and “Fall” confusion. These factors will be further addressed in Section 5 of this paper. Further, the slight differentiation of complex, static activities such as Laptop Lap and Laptop Desk are challenging to identify with the MI-PIR alone. The *PRF-PIR* framework for sensor fusion has been shown to effectively reduce the defects of the MI-PIR system in vertical FoV and ambient dependence. Compared with the MI-PIR system only, the accuracy of the *PRF-PIR* system improved the results by 19.30%.

*PRF-PIR* has achieved promising results in classification accuracy, and the complexity of the *PRF-PIR* system needs to be further analyzed and demonstrated. For the space complexity, the *PRF-PIR* sensor fusion can be regarded as approximately twice that of the single-modal sensor, which can be seen from the number of features. The space complexity depends on our prior assumption of multi-modal sensor fusion, i.e., we assume that both modal sensors have the same contribution to the model. Therefore, PRF and PIR have a similar number of features set in the system. The feature extraction capability of RNN can assign different weights to different features, which is our a posteriori conclusion. For the time complexity, both RNN models trained on the single-modal data set are about 17 s, while the RNN model trained on the multi-modal data set is about 24 s. It is not difficult to see that, compared with MI-PIR, the PRF-PIR system improves the accuracy by 19.70% with little increase in complexity, which can confirm the potential of the proposed sensor fusion system for real-time applications.

### 4.4. Explainable AI (XAI)

While the sensor fusion of the proposed *PRF-PIR* solution performed exceptionally well, due to the nature of any black-box algorithm, it is necessary to explore the predictability of the fusion result further. To combat this, the use of a game-theoretic approach SHAP [53,54] and its related extension is utilized. The decision-level fusion model shows the Shapley values of the above three data sets in different activities, which is the average impact on model output. While there is a fundamental limitation with the FoV of MI-PIR and potential sources of PRF interference, the influence of the above three data sets on the RNN model is still unknown. Shapley values of PRF, PIR, and *PRF-PIR* sensor fusion can reflect these three types of data on the model output impact, which has received our attention.

A linear machine learning technology SVM model was used to train the decision-level data set, and an accuracy of 0.9764 was obtained. The function of this SVM determines the relationship between the input (decision-level data) and the output (predict), or in other words, the interpretability of how the data impacts the model output. For the antenna at position C with less interference, PRF data has a higher impact than PIR, as shown in Figure 7a. The result is the opposite, however, for the situation when the antenna is in the E location and the interference is greater. This result is presented in Figure 7b. Because SDR receives interference of different intensities, sensor fusion can automatically and reasonably perform feature extraction and find a reasonable solution. XAI shows the process of feature extraction and the impact of each activity on the model. Although the *PRF-PIR* sensor fusion system has the highest impact regardless of whether it is in the C or E position, for some specific activities, the sensor fusion system will be affected by single-type sensor data, and the accuracy will decrease.

To obtain the details of a single feature impact on the model, the “Unoccupied” activity is an adequate example to show the Shapley values distribution of all *PRF-PIR* sensor fusion features and the impact of PRF and PIR on the sensor fusion when the antenna is in the C position. The interaction dependence plots of *PRF-PIR* sensor fusion with PRF and PIR are shown in Figure 8a,b, respectively. The interaction dependence plot is used as a three-dimensional point cloud to show the distribution of points (each sample) on the x-axis (sensor fusion activity index), y-axis (Shapley value), and z-axis (single-type sensor activity index). In Figure 8, as an example of “Unoccupied” activity, the points distributed in the exemplified sensor fusion activity index (“Unoccupied” is Activity 0) should have a higher Shapley value, which means sensor fusion has higher accuracy. At the same time, the distribution in the exemplified sensor fusion activity index 0 should have a lower value (color in blue), which will verify the accuracy of the single-type sensor. With the activity index increase, the negative growth of Shapley values should be the trend, as well as the value change (from blue to red). Red and blue arrows in Figure 8 represent the samples that produce typical misjudgments in activities of “Unoccupied” and “Fall" activity, respectively, which shows that PIR samples have more errors than PRF.

The interaction dependence plots indicate a much more substantial impact on the PRF data than the PIR data with the *PRF-PIR* sensor fusion. The use of the interaction dependence plots provides an intuitive and causal explanation for the impact of the two modalities. The usage of the plots has the advantage of showing the distribution of features and how a single feature affects the model.

## 5. Discussion

Our proposed *PRF-PIR* sensor fusion system for an at-home human monitoring solution has proven to be highly accurate, reliable, and robust to HIAR classification tasks, as SHAP XAI has confirmed the feasibility of sensor fusion. In this paper, we highlight two contributions: passive sensor fusion systems and interpretability. Unlike PRF techniques which are wrongly mentioned in [56,57], a truly PRF system does not require RFID tags or any transmitter, but only a receiver passively receives PRF data. Compared with other active RF system, the PRF system mentioned in this paper saves energy, enables custom frequency bands, increases the sampling speed of data, saves computing resources, and reduces the radiation risk of the transmitter. To our best knowledge, our proposed *PRF-PIR* system is the first passive sensor fusion system for human monitoring. Additionally, we introduce interpretability into the system due to the perturbation susceptibility of passive techniques and the black-box property of RNN.

It is well-known that PRF signals are prone to electronic interference, which comes from electronic products such as mobile phones, computers, refrigerators, and reflection from surrounding objects, especially metal objects. Although [35] found human-sensitive frequency bands circumvent the radio frequency of electronic products, the interference caused by metal cannot be avoided. The unpredictability and randomness of this kind of metal object interference bring troubles to the PRF signal; however, our *PRF-PIR* sensor fusion system can avoid the interference of metal objects to a great extent due to its multiple data acquisition sources.

Deployed in the traditional sense, PIR sensors are dependent on their ambient environment as changes in infrared radiation can cause false positives such as with changes in illumination or other objects in motion, i.e., ball, animal, drones, or robotic vacuums. While rotating a PIR sensor such as in the MI-PIR system, this dependence on the ambient environment is significantly heightened. Due to this dependence, sensor fusion was hypothesized to increase classification accuracy and robustness. This can be seen with the increased accuracy from a single MI-PIR system to the *PRF-PIR* system. Also, this dependence on the ambient environment and limited vertical FoV is illuminated with the simulated fall activity. The human subject lies on the floor of the indoor environment, much lower than the cabinet that the MI-PIR system rests on. In addition, one of the two unoccupied labels was collected on the same day as that of all of the simulated fall events. Due to the ambient environment being very similar on the same day, as well as the human subject lying much lower than the height of the MI-PIR system, there exists quite a bit of confusion about the RNN model.

The inclusion of the SDR device to the MI-PIR system extends the work presented in [18], and allows for a more accurate at-home monitoring solution where both the ambient dependence and sensor locations will be altered based on the home of the user. A long-term solution for the MI-PIR system would be to reduce its size and place at a high vantage point, further reducing intrusion to the end-user. In this future implementation, the SDR device has been shown to reduce inaccuracies resulting from the limited vertical FoV. Overall, the *PRF-PIR* sensor fusion system combatted the vertical FoV and ambient dependence shortcomings of the MI-PIR system.

## 6. Conclusions

In this work, we propose *PRF-PIR*, an interpretable, passive, multi-modal, sensor fusion framework for human monitoring to classify HIAR tasks via an RNN model architecture with LSTM units. The proposed *PRF-PIR* system comprises one SDR device and one MI-PIR system. The proposed sensor fusion model proves effective at decreasing the effects caused by external interference on an SDR device antenna and the effects caused by limited vertical FoV and ambient dependence on the MI-PIR system. The *PRF-PIR* sensor fusion model accuracy is based on the optimal placement of the SDR device antenna, which was determined experimentally by the strategic placement of five antennas during the data collection period. Overall, the *PRF-PIR* sensor fusion model proved 0.9866 at human identification of twelve human subjects performing the same smartphone static activity and 0.9623 at activity recognition of eleven activities performed by twelve human subjects. These reported accuracies are a maximum 9.70% and 19.30% accuracy increase over the single modality solutions in the optimal antenna location for human identification and activity recognition, respectively. The sensor fusion accuracies prove to be a 21.52% and 23.73% accuracy increase over the SDR device Antenna E, which received the most impact from the external interference.

To further illustrate the robustness of the *PRF-PIR* sensor fusion system, the decision-level fusion XAI SHAP is used to interpret how to reduce the disadvantage of PIR in recognition of specific activities and the interference of PRF in different antenna locations. After summarizing the impact of twelve activities on the model output, the interaction dependence plot demonstrates more details of the impact of the feature. The decision-level fusion XAI SHAP has given us the ability to understand the sensor fusion process and the application prospects of visualization. In conclusion, the *PRF-PIR* proposed sensor fusion model provides an adequate solution that leverages passive and non-intrusive sensing devices that make for an advantageous human monitoring system.

To improve the *PRF-PIR* sensor fusion system for human monitoring, our future work is indicated below. The first future work of high importance is the joint data collection of subjects and the data collection of actual falls in a real-world scenario. This will assist in the validation of the *PRF-PIR* system. In addition, enhancements to the MI-PIR system, such as reducing the size, altering the location of the sensor system for an end-user, and exchanging the thermal insulation material, will be undertaken.

Beyond human monitoring future work, the *PRF-PIR* sensor fusion system could be useful for extended applications. Thus, future work includes testing the sensor fusion system in larger application areas, such as classrooms, basketball courts, etc. Additionally, testing the sensor fusion solution in more common environments, such as vehicles, may be of benefit and is considered for future work. In addition, multi-subject scenarios will be addressed as achieving HIAR for multiple subjects at the same time is a more realistic task.

## Figures and Tables

**Figure 1 sensors-22-05787-f001:**
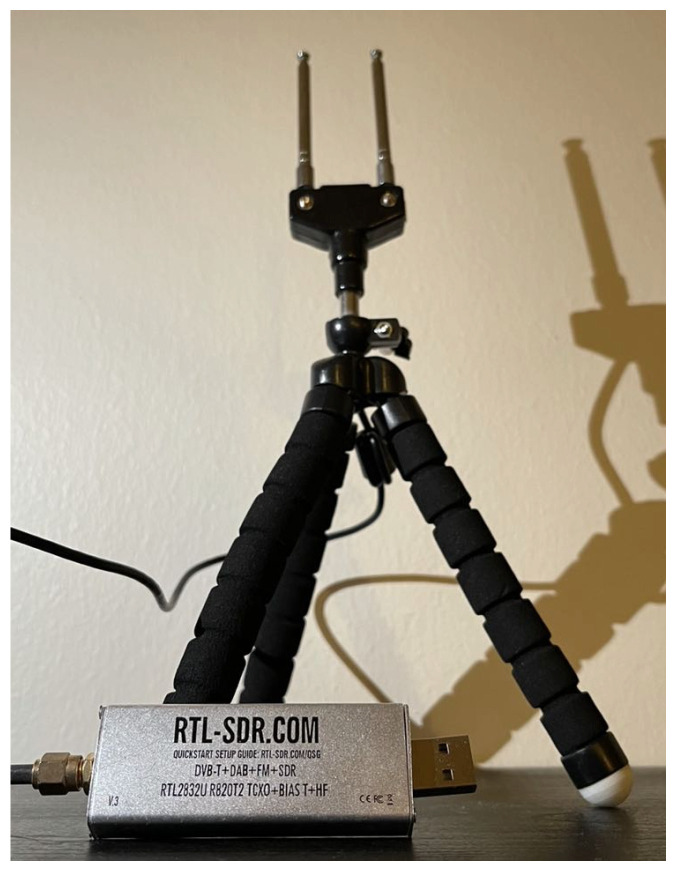
SDR RTL2832U and antenna used in PRF data collection.

**Figure 2 sensors-22-05787-f002:**
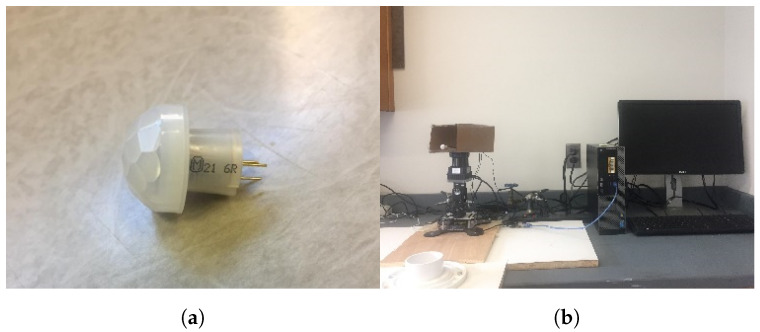
(**a**) The Panasonic AMN 24112 PIR sensor and (**b**) the complete MI-PIR system.

**Figure 3 sensors-22-05787-f003:**
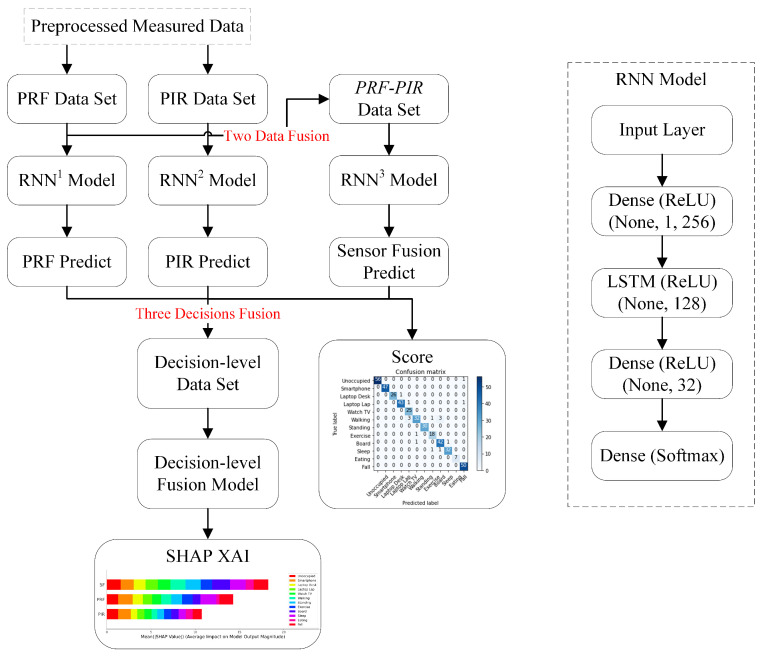
System diagram of *PRF-PIR* sensor fusion system. RNN${}^{1}$, RNN${}^{2}$, and RNN${}^{3}$ use the same model architecture and are used to train different data sets to obtain different models. The *PRF-PIR* data set is used to train RNN${}^{3}$ to obtain sensor fusion prediction. Two data fusion and three decisions fusion are marked in the system diagram.

**Figure 4 sensors-22-05787-f004:**
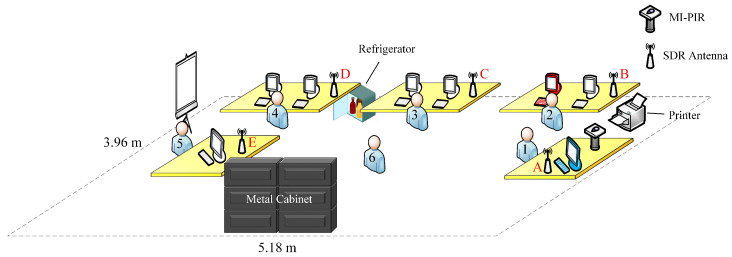
Illustration of the academic office space used to collect the data. The red and blue computers are used as the host computers for the SDR devices and the MI-PIR system, respectively. The white computers are not used.

**Figure 5 sensors-22-05787-f005:**
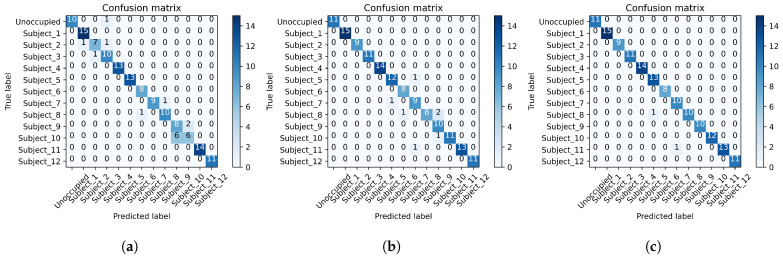
Confusion matrices for the human identification results of (**a**) PRF, (**b**) PIR, and (**c**) *PRF-PIR* sensor fusion.

**Figure 6 sensors-22-05787-f006:**
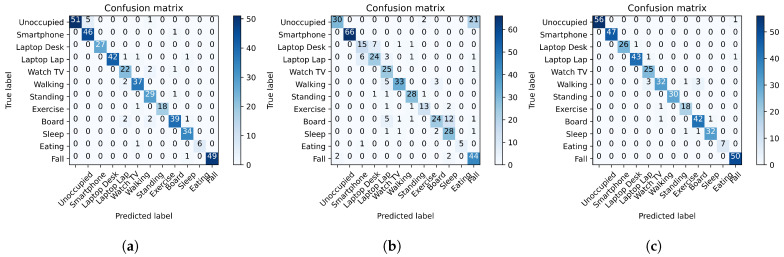
Confusion matrices for the activity recognition results of (**a**) PRF, (**b**) PIR, and (**c**) *PRF-PIR* sensor fusion.

**Figure 7 sensors-22-05787-f007:**
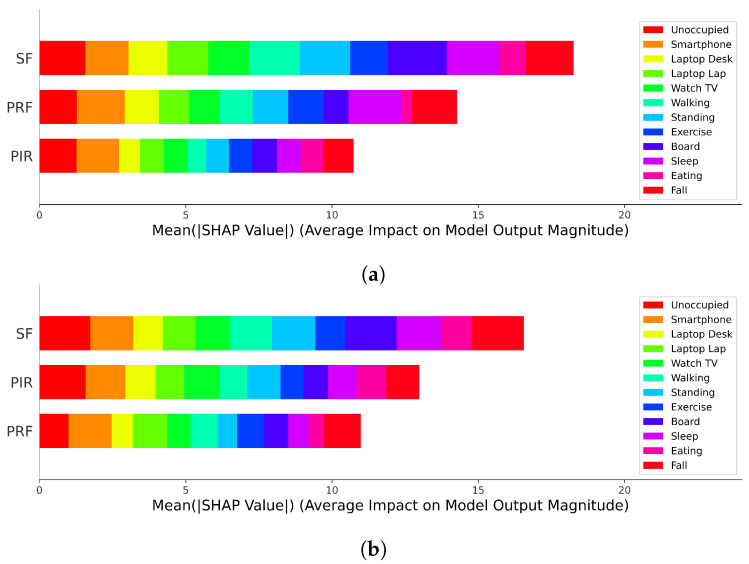
Summary plot of PRF, PIR, and *PRF-PIR* sensor fusion Shapley values when the antenna is in (**a**) C location and (**b**) E location.

**Figure 8 sensors-22-05787-f008:**
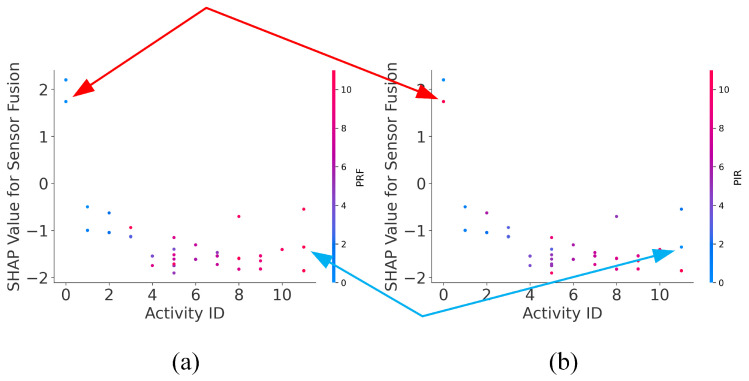
Interaction dependence plot of (**a**) *PRF-PIR* with PRF and (**b**) *PRF-PIR* with PIR.

**Table 1 sensors-22-05787-t001:** Data collection table including activity, human subject, location, and description.

Activity Recognition ID	Activity	Human Subject ID	Location	Description
0	Unoccupied	0	N/A	Unoccupied.
1	Smartphone	1-12	6	Subject sits and uses a smartphone.
2	Laptop Desk	2-3	2-4	Subject sits and uses a laptop placed on the desk.
3	Laptop Lap	1-2,4,7	6	Subject sits and uses a laptop placed on the lap.
4	Watch TV	1-3	4,6	Subject sits and watches videos on a laptop placed on a desk.
5	Walking	1-4	Walk	Subject walks randomly in the laboratory.
6	Standing	1-4	4-6	Subject stands.
7	Exercise	2-3	6	Subject performs random exercises continuously.
8	Board	1-4	5	Subject writes on a whiteboard.
9	Sleep	2-5	3	Subject placed their head on a desk.
10	Eating	2	4	Subject eats at a desk.
11	Fall	1-4	6	Subject lies on the ground to simulate a fall.

**Table 2 sensors-22-05787-t002:** Differences in subjects’ age, weight, height, and BMI information.

	Minimum	Maximum	Average	Standard Deviation	Variance
Age (years old)	20.00	27.00	22.75	2.34	5.48
Weight (kg)	51.00	100.00	70.25	13.10	171.66
Height (cm)	162.00	188.00	176.17	8.04	64.70
BMI	17.65	29.86	22.55	3.36	11.29

**Table 3 sensors-22-05787-t003:** Comparison between sensor fusion and SDR only in different locations.

Activity Recognition				
**Antenna Location**	**PIR**	**PRF**	**SF**	**SF vs. PRF Improvement** ^1^
A	0.8066	0.9080	0.9458	4.16%
B		0.8868	0.9575	7.98%
C		0.9434	0.9623	2.00%
D		0.8915	0.9646	8.20%
E		0.7453	0.9057	21.52%
**Human Identification**				
**Antenna Location**	**PIR**	**PRF**	**SF**	**SF vs. PRF Improvement** ^1^
A	0.9530	0.8389	0.9664	15.20%
B		0.8054	0.9866	22.50%
C		0.8993	0.9866	9.70%
D		0.8456	0.9799	15.87%
E		0.7919	0.9799	23.73%

^1^ Sensor fusion (SF) vs. PRF Improvement is the relative improvement of sensor fusion compared to PRF.

## Data Availability

Not applicable.

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
