# Peer review of "Interpretable Passive Multi-Modal Sensor Fusion for Human Identification and Activity Recognition"

_sensors, 2022, doi:10.3390/s22155787_

Round 1

Reviewer 1 Report

This paper presents an interpretable passive multi-modal sensor fusion system to address the shortcomings of single modality sensor systems. The proposed method is demonstrated on a potential human monitoring system through the data collection of eleven activities from twelve human subjects in an academic office environment.

Some explanations on sensor fusion at the data level should be given. How is this done?

It is mentioned in Section 3.2 that “the PRF, PIR, and PRF-PIR data sets are trained by the same RNN model architecture”. Do you mean the same RNN model is developed on these 3 data sets or 3 RNN models with the same structure are developed on the three data sets respectively? Some clarifications should be given.

There is no mention of data partition into training, testing, and validation data sets. The generalisation capability of the developed models is of more importance and this needs to be demonstrated on a set of unseen data. Did you use all the data to train the RNN models? Are the results given in Table 3 on the training data?

Reviewer 2 Report

This work proposes a PRF-PIR system, an interpretable passive multi-modal sensor fusion framework for elderly monitoring, to classify HIAR tasks via an RNN model architecture with LSTM units. The proposed sensor fusion model effectively decreases the effects caused by external interference on an SDR device antenna and the effects caused by limited vertical FoV and ambient dependence on the MI-PIR system. This work is exciting and can be helpful for many applications. After reading the manuscript, here are my comments:

  1. The work was conducted only with twelve different human subjects. The data collection process was completed in an indoor office space on the campus of Oakland University. How can we be sure that the results from this number of experimental groups can be used to develop a care system for the elderly?
  2. The PRF-PIR system has been shown to effectively reduce the defects of the MI-PIR system in vertical FoV and ambient dependence. Compared with the MI-PIR system only, the accuracy of the PRF-PIR system improved the results by 19.30%. How about the complexity? Please show the complexity analysis of the PRF-PIR system.
  3. Please discuss more details in Antenna Location C with the derived least accuracy.
  4. I think it is not fair to compare the results with the state-of-the-art studies since you did not include the data of 12 participants. Please explain this.
  5. Please define "Location 1" in line 365.
  6. Many abbreviations have not been defined in their appearance, such as HIAR.
  7. "This interpretable passive multi-modal sensor fusion solution," please consider the noun phrase.

I consider that the authors need to address these issues before their paper can be published in the journal.

Reviewer 3 Report

  1. Explainable AI: The authors mention interpretability as one of their contributions. LIME (Local Interpretable Model-Agnostic Explanations) are often used in deep learning experiments in healthcare contexts including involving RNN models. LIME, which uses perturbed sampling for creating explanations, when used along with SHAP can provide much more understanding of the model than using SHAP alone. Further, alongside using aggregate SHAP values across the entire dataset, the authors should choose a handful of data-points including points which were incorrectly predicted and demonstrate the SHAP values across classes for that point. 

  1. The data does not seem representative, and information about gender is missing and this is correlated with height and weight. The average US person has a BMI of around 26.6 and the study participants have average of 22.55. It is also not clear if the postures of senior citizens when they are unoccupied, walking or exercising will match that of college students. The authors should consider if they still want to emphasize about elderly monitoring in the introduction. 

  1. “For decision level fusion model itself, the use of SVM, a linear supervised learning methods adds further transparency”. This is not really true as in traditional machine learning (outside of deep learning), SVMs are notorious for being difficult to understand. Also, SVMs are not necessarily used for linear problems given the use of kernels allows them to solve non-linear problems. The authors should consider using SHAP values to explain SVM. 

  1. It would be useful to see a top k-accuracy score and see the 2nd most probable class especially for Figure 6(b) regarding the error caused due to “Unoccupied” vs. “Fall” confusion. If the model is being trained well, the 2nd most probable class would still match with the correct label even if the most probable class is not. 

  1. There are some confusing statements in the paper.

    1.  “Unlike PRF techniques which are wrongly mentioned in [58,59]”- the use of wrongly is very confusing and should be better explained. 

  1. “Four machine learning models, support vector machine (SVM), k-nearest neighbors (k-NN) and linear SVM with SGD” is confusing. What is the difference between first mention of SVM vs the linear SVM with SGD ? Does the first SVM uses a polynomial or rbf kernel as opposed to a linear SVM ?

  1. The mention of population increase in relation to fall events in the introduction does not seem relevant and confusing as population in many developed countries including the US may fall in near future. Generally, it is enough to mention that the ratio of senior citizens to working adult population would increase. The mention of population makes the readability of the paper low, and if population is mentioned, it should be mentioned in detailed with proper rationale. 

  2. “Deep learning in comparison to machine learning” is confusing and potentially incorrect. Deep learning is the use of deep (many layers of) neural networks which in turn are a machine learning algorithm. As deep learning is a subset of machine learning, it is more correct to say deep learning in comparison to other machine learning methods or traditional ML.

Round 2

Reviewer 1 Report

The authors have adequately addressed my comments and the revised manuscript can be accepted.

Reviewer 2 Report

In this version, the authors have added some experimental results and modifications to respond positively to my questions.